# Using canavanine resistance to measure mutation rates in *Schizosaccharomyces pombe*

**Chen-Chun Pai**[1]*, **Ellen Heitzer**[2], **Sibyl Bertrand**[3], **Sophia Toumazou**[3], **Timothy C. Humphrey**[1], **Stephen E. Kearsey**[3]*

**1** Department of Oncology, CRUK-MRC Oxford Institute for Radiation Oncology, University of Oxford, ORCRB, Oxford, United Kingdom, **2** Institute of Human Genetics, Diagnostic & Research Center for Molecular BioMedicine, Medical University of Graz, Graz, Austria, **3** ZRAB, University of Oxford, Oxford, United Kingdom

* stephen.kearsey@biology.ox.ac.uk (SEK); chen-chun.pai@eng.ox.ac.uk (C-CP)

**Data Availability Statement:** All sequencing data are available now at the NCBI-SRA database, accession number PRJNA877936 (https://www.ncbi.nlm.nih.gov/sra/PRJNA877936).

## Abstract

We constructed a panel of *S. pombe* strains expressing DNA polymerase ε variants associated with cancer, specifically POLES297F, POLEV411L, POLEL424V, POLES459F, and used these to compare mutation rates determined by canavanine resistance with other selective methods. Canavanine-resistance mutation rates are broadly similar to those seen with reversion of the *ade-485* mutation to adenine prototrophy, but lower than 5-fluoroorotic acid (FOA)-resistance rates (inactivation of $ura4^+$ or $ura5^+$ genes). Inactivation of several genes has been associated with canavanine resistance in *S. pombe* but surprisingly whole genome sequencing showed that 8/8 spontaneous canavanine-resistant mutants have an *R175C* mutation in the *any1/arn1* gene. This gene encodes an α-arrestin-like protein involved in mediating Pub1 ubiquitylation of target proteins, and the phenotypic resistance to canavanine by this single mutation is similar to that shown by the original "*can1-1*" strain, which also has the *any1R175C* mutation. Some of the spontaneous mutants have additional mutations in arginine transporters, suggesting that this may marginally increase resistance to canavanine. The *any1R175C* strain showed internalisation of the Cat1 arginine transporter as previously reported, explaining the canavanine-resistance phenotype.

## Introduction

Measuring mutation rates using resistance to the toxic amino acid canavanine has been widely used in fluctuation analysis in *Saccharomyces cerevisiae* [1–3]. In this organism, the basis of the resistance is simple in that mutation of the arginine transporter *CAN1* blocks canavanine import, and sequencing of mutations in the *CAN1* gene can be used to determine the mutational spectra of different yeast strains [4]. In *Schizosaccharomyces pombe*, canavanine resistance has also been used in mutation rate analysis [5–7], but here the genetic basis of the resistance is less clear. An early study of a "*can1-1*" strain, identified *can1* and *canX* as responsible for the canavanine resistance phenotype [8], but recent studies showed that the phenotype is due at least in part to an R175C missense mutation in the *any1* gene, which is closely linked to $can1^+$(now renamed $vhc1^+$) [9,10]. However, loss of function mutations in a number

**Funding:** This work was funded by the Medical Research Council (grant MR/L016591/1 to S.E.K.) The funders had no role in study design, data collection and analysis, decision to publish, or preparation of the manuscript.

**Competing interests:** The authors have declared that no competing interests exist.

of *S. pombe* genes, such as the arginine-transporter *cat1*[+], the cyclin *pas1*[+] [11] and the *tsc1*[+]/*tsc2*[+] genes involved in TORC1 signalling [12,13] have also been reported to confer canavanine resistance, so it is not clear which genes are most commonly mutated in spontaneous canavanine-resistant mutants generated in fluctuation analysis experiments. This information would be useful as it would facilitate directed sequencing to determine mutation spectra and would clarify whether mutation rates determined are likely to be representative of genome-wide mutation rates.

We present here a simple protocol for carrying out mutation rate assays using fluctuation analysis in *S. pombe*. We generated a panel of *S. pombe* strains expressing DNA polymerase ε variants with mutations in the exonuclease domain of Pol2, which have been identified as cancer-associated variants, and compared mutation rates using canavanine resistance, reversion of an *ade6-485* auxotrophic mutation, and resistance to 5-fluoroorotic acid (FOA) generated by loss-of-function mutations in *ura4*[+] or *ura5*[+] genes. The genomes of eight canavanine-resistant mutants were sequenced and surprisingly all had an *any1R175C* mutation. Comparison with strains with single mutations in amino-acid transporters showed that the *any1R175C* mutation confers a more pronounced resistance to canavanine.

## Materials and methods

### *S. pombe* methods

Standard media and genetic techniques were used as previously described [14]. Cultures were grown in either rich medium (YE3S), EMM or PMG plates supplemented with the appropriate supplements and incubated at 30˚C. Nitrogen starvation was carried out using EMM lacking NH₄Cl. The relevant genotypes and source of the strains used in this study are listed in S1 Table. Oligos used for strain constructions and genotyping are listed in S2 Table.

Spontaneous canavanine-resistant mutants were derived by setting up separate cultures starting at $10^4$ cells/ml, growing to saturation, and then spotting 150 ul of each culture onto PMG+canavanine plates (80μg/ml) and incubating for 1 week at 30˚C. Only one canavanine-resistant colony per spot was selected. The level of canavanine-resistance was determined by spot testing, and strains were only selected for sequencing if resistance was comparable to the original "*can1-1*" strain (3647).

For spot testing, cells were grown overnight to exponential phase and serial dilutions (1/10) of cells were spotted on agar plates using a replica plater (Sigma R2383). Plates were photographed after 2–5 days at 30˚C.

An AxioPlan 2 microscope was used for fluorescence microscopy as previously described [5] and pictures were taken using a Hamamatsu ORCA E camera system with Micro-Manager 1.3 software.

### Construction of *S. pombe* strains

The *pol2L425V* strain was constructed by amplifying *pol2*[+] segments with primer pairs; 1083 and 1108 and 1084 and 1109. The products were purified, annealed and amplified using primers 1083 and 1084. The product was digested with BglII and AscI and inserted into BamHI, AscI-cleaved pFA6a-*kanMX6* [15]. The *pol2V412L* strain was similarly constructed using primer combinations:1083 and1137, 1084 and 1138; and for the *pol2S298F* strain the oligos 1083 and 1144, 1084 and 1145 were used. Plasmids were integrated into the *pol2*[+] locus after linearization with BamHI and selecting for G418 resistance. Constructs were verified by Sanger sequencing.

The *vhc1Δ*::*hphMX6* strain was constructed using oligos 1382 and 1383, using plasmid pFA6ahphMX6 as template. *Vhc1Δ* strains were identified using colony PCR with oligos 1432 and 1433.

Strains were genotyped for the *any1R175C* mutation by colony PCR (94˚C 5 min, then 30 cycles of 94˚C 30 sec, 62˚C 15 sec, 68˚C 15 sec) using oligos 1420 and 1421, which gives a product with the wild-type strain, and oligos 1419 and 1421 which gives a product with the *any1R175C* strain (product size = 600 bp). Alternatively, oligos 1436 and 1437 were used to amplify a segment of the *any1* gene, and this was subsequently digested with HpyCH4IV. The *any1R175C* mutation destroys a HpyCH4IV site, and gives fragment sizes of 373 and 289 bp, whereas the WT gene gives fragment sizes of 289, 260 and 113 bp.

## Determination of mutation rates by fluctuation analysis

The protocol for mutation rate assays was derived from [2]. Briefly, for each assay a culture ($10^4$ cells/ml) was aliquoted into 12 wells of a 96-well microtitre plate (0.25 ml/well) and allowed to grow to saturation. Dilutions were plated onto YES plates to determine the cell concentration and, for canavanine-resistance, 0.15 ml of each culture was plated out onto predried PMG plates (EMM-G, [8]) containing 80 μg/ml canavanine. We found that reducing the pH of the PMG medium to 4.5 reduced the time necessary to count canavanine-resistant colonies (S1 Fig), which were counted after 5–10 days at 30˚C. For FOA resistance, cells were plated onto EMM medium containing 1 g/l FOA (Formedium). Since background growth on the primary FOA-containing plates made it difficult to score FOA-resistant colonies, the primary plates were replica plated onto fresh FOA media after 3 days and scored after an additional 3 days. For mutation rate determination based on reversion of the *ade6-485* allele, cells were plated onto EMM-adenine plates. Each assay was repeated at least three times from independent clones.

Mutation rates were calculated using the Ma-Sandri-Sarkar maximum likelihood estimator [16], implemented in rSalvador [17].

## Whole genome DNA sequencing

*S. pombe* DNA was extracted from cells grown to log phase at 32˚C using a YeaStar genomic DNA purification kit (Zymo Research). Equal quantities of DNA from four spontaneous canavanine-resistant clones derived from strain 2299 were combined; similarly, DNA from four canavanine-resistant clones from strain 3938 were combined. DNA from strain 3647 ("*can1-1*") was sequenced as a unique sample. Genomic DNA sent to Novogen (Cambridge, UK) Company Limited for whole genome sequencing (Illumina PE150). The resulting reads were aligned to the reference genome *Schizosaccharomyces pombe* ASM294v2 using BWA software (parameters: mem -t 4 -k 32 -M). Duplicates were removed by SAMTOOLS. The average sequencing depth was 68–107. Variants were called using GATK HaplotypeCaller. Sequence data are available from the NCBI-SRA database:

https://www.ncbi.nlm.nih.gov/sra/PRJNA877936.

## Results and discussion

We generated a panel of *S. pombe* strains expressing DNA polymerase ε variants with mutations in the exonuclease domain of the largest subunit Pol2/Cdc20 (Table 1, Fig 1A). These variants are either germline or somatic mutations associated with predisposition to colorectal and endometrial cancers [18]. In addition, we used an *S. pombe* Pol2D276AE278A mutant where the 3'-5' exonuclease activity required for proofreading is inactivated. Mutation rates of these strains were determined by fluctuation analysis using canavanine resistance. In parallel, mutation rate assays were carried out selecting for Ade$^+$ reversion of the *ade6-485* allele, which requires acquisition of missense mutation in a stop codon, and selection for FOA resistance, which is caused by mutations inactivating the *ura4$^+$* or *ura5$^+$* genes.

**Table 1. DNA polymerase ε variants analysed in Fig 1B.**

| POLE clinical variant | S. pombe equivalent mutation in Pol2 | Germline/somatic | Reference |
|---|---|---|---|
| - | D276A E278A | (no 3'-5' exo activity) | [5] |
| S297F | S298F | Somatic | [20] |
| V411L | V412L | Somatic/Germline | [20,21] |
| L424V | L425V | Somatic/Germline | [22] |
| S459F | S460F | Somatic | [18,23] |

The mutation rates determined by canavanine-resistance were lower than those determined by FOA resistance but showed a similar trend, and were quite comparable to those determined by *ade6-485* reversion (Fig 1B). Interestingly two of the variants, S298F and S460F showed hypermutation, which we define as having a mutation rate higher than that of a strain with simple loss of exonuclease activity (i.e. Pol2D276AE278A). This suggests that the mechanism of mutagenesis must involve something other than a defect in proofreading, which has also been shown for a yeast strain expressing the equivalent mutation to the POLE P286R variant [5,24]. Also of interest is that the V412L variant shows a mutation rate lower than the Pol2-D276AE278A strain, while the equivalent human mutation V411L causes hypermutation [20]. A similar low mutation rate was shown using *S. cerevisiae* to model the mutation [25], suggesting that this amino-acid substitution may affect the yeast and human polymerases differently.

To determine the genes mutated in the canavanine fluctuation analysis, the genomes of eight canavanine-resistant mutants were sequenced. Mutants were selected only if their canavanine-resistant phenotype was at least as pronounced as the original "*can1-1*" mutant (3647). We initially sequenced the *vhc1⁺* gene, previously called *can1⁺*, of the canavanine-resistant mutants by Sanger sequencing, but none showed a mutation. Whole genome sequencing showed that surprisingly all eight mutants had the *any1R175C* mutation (Table 2), recently also identified in the "*can1-1*" mutant [9,10]. Three canavanine-resistant strains additionally had a missense mutation in the transmembrane transporters *cat1* or *aat1*. These mutations affected residues that are conserved between Cat1 and Aat1 (S2 Fig). We additionally sequenced the original "*can1-1*" strain and this showed the *any1R175C* mutation as recently reported [9,10]. In addition, there was a frameshift mutation in the cyclin *pas1⁺* gene. Since inactivation of this gene has been shown to cause canavanine resistance [11], this may contribute to the phenotype of this strain, and may correspond to '*canX*' in the analysis of Fantes and Creanor [8]. We also found that independent canavanine-resistant clones from the hypermutating strain *pol2S298F* had the *any1R175C* mutation (S3 Fig).

The R175C mutation occurs in the arrestin-fold domain of Any1 (Fig 2), which interacts with transmembrane transporters such as Aat1 and probably Cat1, allowing ubiquitylation by the Pub1 ubiquitin ligase [26,27]. R175 is conserved in the *S. pombe* Any1 homologue Any2/Arn2 (Fig 2), where it is close to K163 which has been identified as a ubiquitylation site [28], and the *S. cerevisiae* orthologue Art1. However the only ubiquitylation site identified in Any1 is K263 [26].

To compare the canavanine resistance of an *anyR175C* strain with mutants defective in amino-acid transporters, we carried out spot assays on PMG+canavanine plates (Fig 3). The *any1R175R* strain showed similar growth to the original "*can1-1*" strain. Strains with single deletions of the *vhc1⁺* (originally *can1⁺*), *cat1⁺*, *aat1⁺* and *SPBPB2B2.01* genes showed no growth on canavanine. A strain deleted for both *aat1⁺* and *cat1⁺* genes showed some growth, although to a slightly lesser extent that with the *any1R175C* strain. Consistent with these findings, Aspuria and Tamanoi [12] referred to unpublished data showing that in the presence of

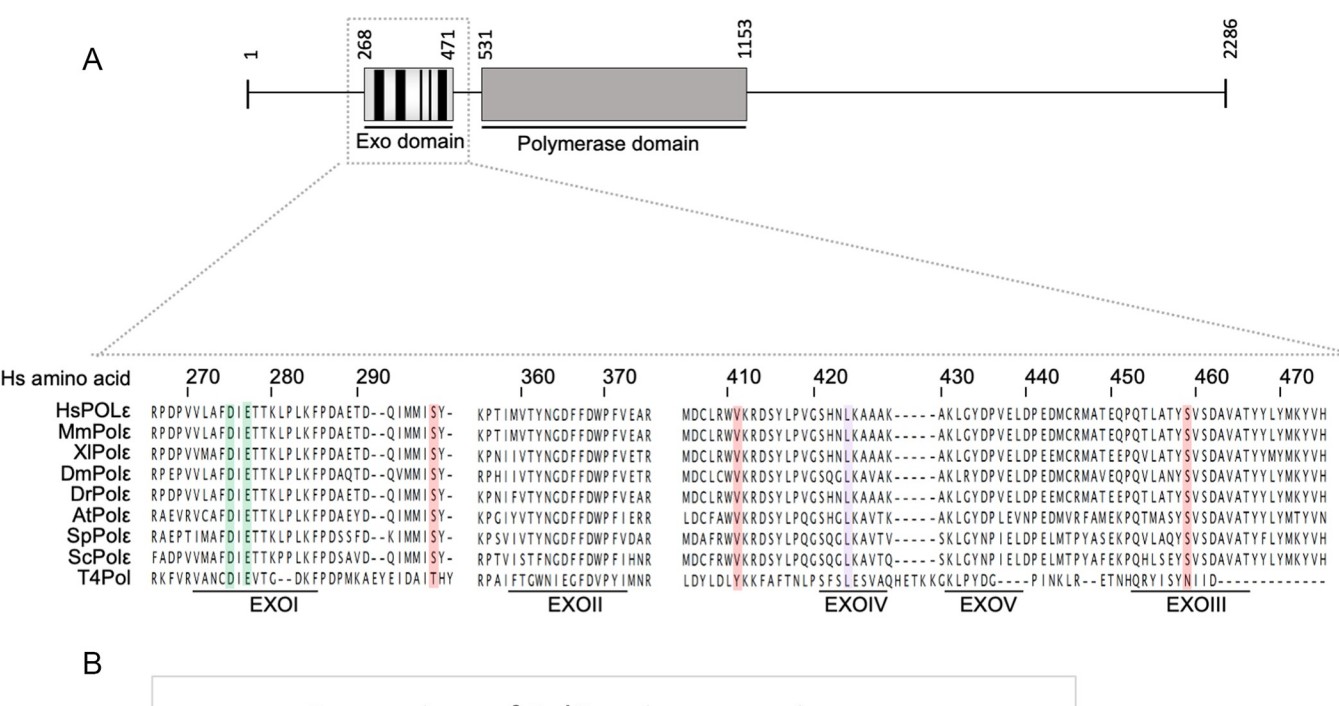

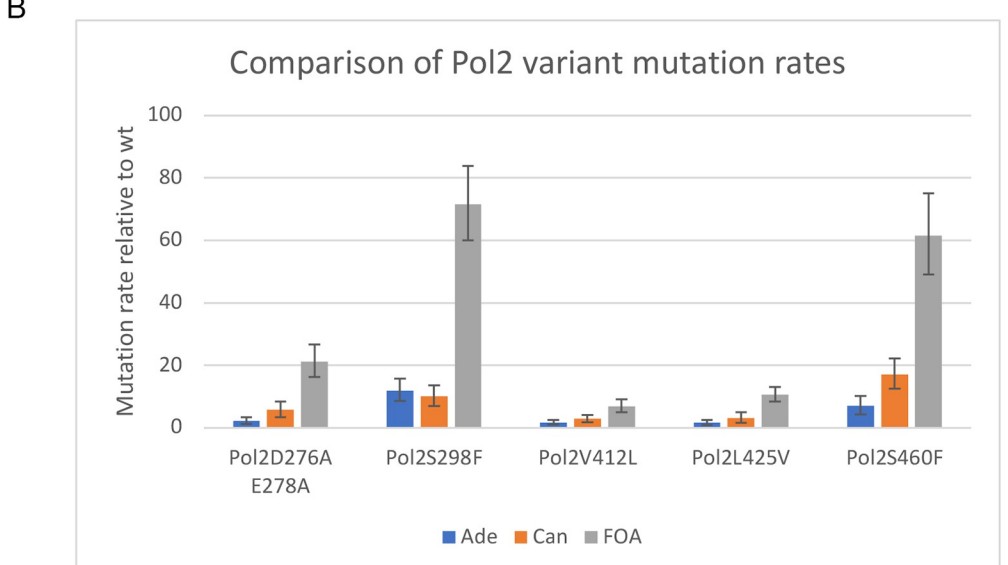

**Fig 1. Mutation rates of DNA polymerase ε variants.** (A) DNA polymerase ε variants used in this study showing alignment of exonuclease domain. Top panel shows a schematic representation of the human catalytic subunit of Pol ε. The 2286 amino acid long subunit contains the two catalytic activities of Pol ε: 3'-5'exonuclease and 5'-3' polymerase activity represented by the exo domain and polymerase domain, respectively. The exonuclease domain possesses five highly conserved domains called Exo motifs (I-V), highlighted in black. Lower panel shows an amino acid alignment of Pol ε and T4 polymerase sequences using T-Coffee [19] (default parameters; no matrix). Somatic and germline/somatic exonuclease domain mutations are highlighted in red, and violet, respectively, while the two catalytic sites are highlighted in green. Further details of the human clinical variants are shown in Table 1. Hs: human; Mm: mouse; Xl: *Xenopus laevis*; Dm: *Drosophilia*; Dr: *Danio rerio*; At: *Arabidopsis thalinana*; Sp: *Schizosaccharomyces pombe*; Sc: *Saccharomyces cerevisiae*; T4Pol: T4 DNA polymerase. (B) Mutation rates determined by fluctuation analysis of five DNA polymerase variants modelled in *S. pombe*, using *ade6-485* reversion, resistance to canavanine or resistance to FOA. Y axis values represent: mutation rate of mutant/mutation rate of wt. Error bars show the range of three independent experiments. Correlation coefficients to the canavanine resistance data are 0.67 (Ade[+] reversion) and 0.86 (FOA resistance). Absolute mutation rates are given in S3 Table.

ammonium Cat1 is the major route for uptake of canavanine but in the absence of ammonium, *cat1Δ* cells are no longer canavanine resistant, but that a double *cat1Δ aat1Δ* mutant shows resistance.

**Table 2. Genes mutated in spontaneous canavanine-resistant mutants.**

| Spontaneous canavanine-resistant strain | Strain number | Derived from | Mutations in genes relevant to canavanine resistance phenotype | |
|---|---|---|---|---|
| Original "can1-1" strain | 3647 | | *any1R175C* | *pas1F63* (frameshift) |
| #1 | 4068 | 2299 | *any1R175C* | *cat1L501R* |
| #2 | 4089 | 2299 | *any1R175C* | |
| #3 | 4090 | 2299 | *any1R175C* | |
| #4 | 4091 | 2299 | *any1R175C* | |
| #5 | 4092 | 3938 | *any1R175C* | *aat1A233S* |
| #6 | 4093 | 3938 | *any1R175C* | *aat1N237K* |
| #7 | 4094 | 3938 | *any1R175C* | |
| #8 | 4095 | 3938 | *any1R175C* | |

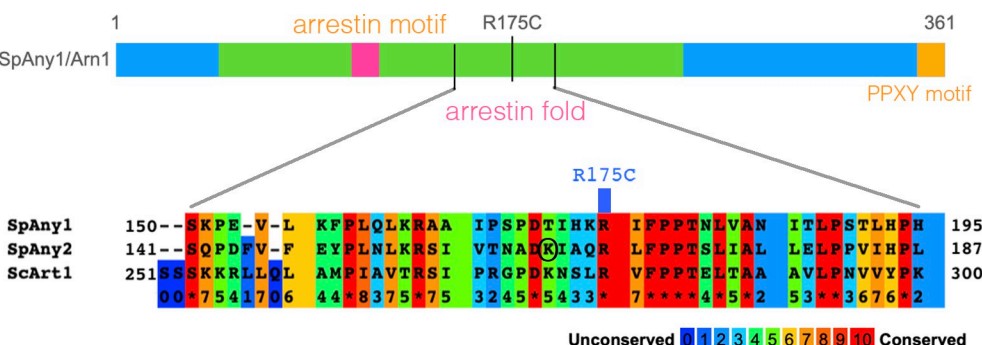

**Fig 2. Location of the R175C mutation in Any1.** The lower panel shows an alignment of *S. pombe* Any1, Any2 and *S. cerevisiae* Art1, generated by Praline [29].

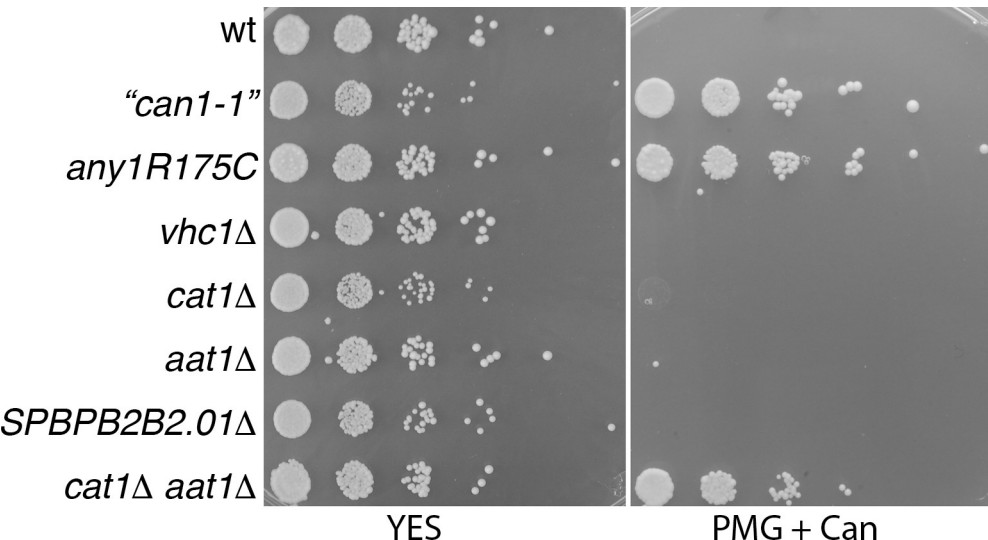

**Fig 3. Comparison of canavanine-resistance phenotype of strains mutated in transmembrane transporters and *vhc1*⁺.** Serial (1/10) dilutions of strains were spotted on the YES and PMG+canavanine plates (80 μg/ml) and incubated at 30°C for 2–5 days. Strains used were: 2299 (wt); 3647 (*can1-1*); 4089 (*any1R175C*); 3791 (*vhc1Δ*); 4059 (*cat1Δ*); 3790 (*aat1Δ*); 3796 (*SPBPB2B2.01Δ*); 3797 (*aat1Δ cat1Δ*).

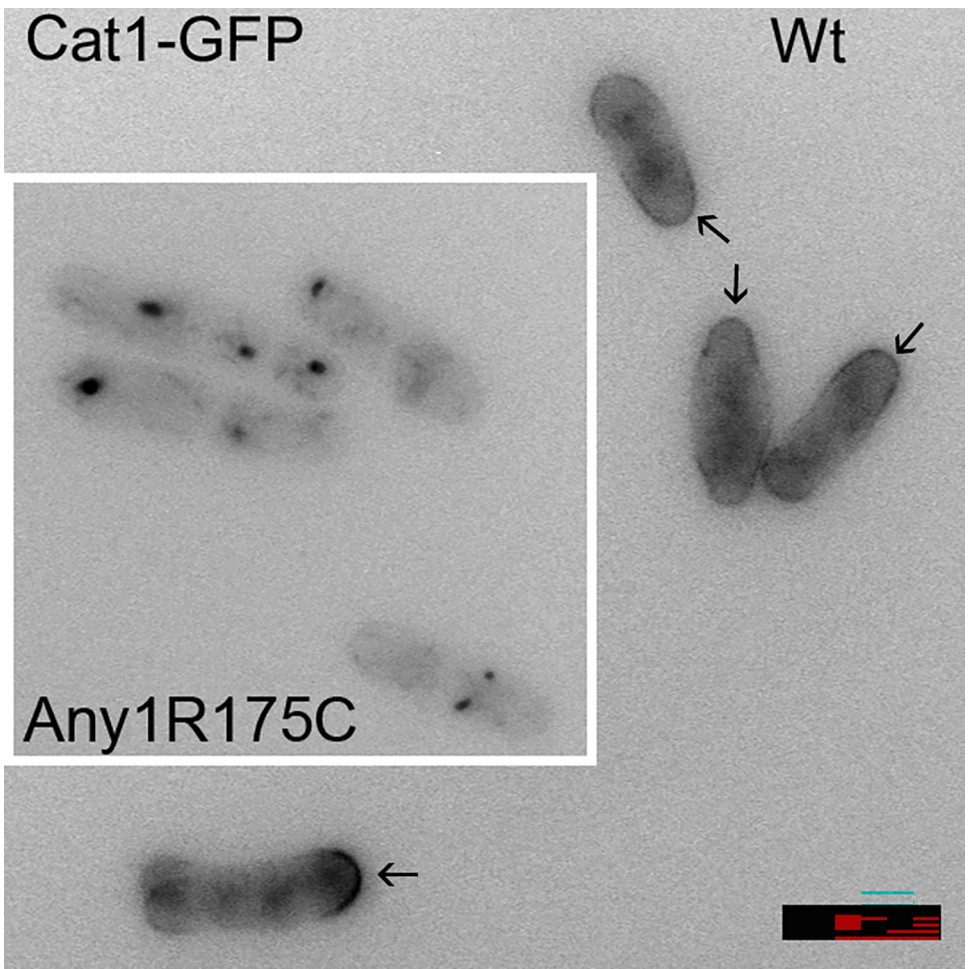

**Fig 4. Cat1-GFP localisation in WT and *any1R175C* cells.** Wild-type cells show Cat1 externalised to the plasma membrane, particularly to the growing tip ends, but in *any1R175C* cells Cat1 is internalised, presumably to the Golgi. Cells were grown to log phase in EMM+NH$_4$Cl, washed and then transferred to EMM-NH$_4$Cl for 60 min before imaging as live cells. Scale bar = 10μm.

### Mechanism of canavanine resistance conferred by Any1R175C

Any1/Arn1 is an arrestin of the HECT-type E3 ubiquitin ligase Pub1 and regulates amino-acid transporters Aat1 and probably Cat1 by ubiquitylation [26,27]. In *S. pombe*, ubiquitination of these amino acid transporters causes them to be transported away from the plasma membrane to the Golgi under nutrient-rich conditions [26,27]. Loss of Any1/Arn1 leads to canavanine sensitivity as Cat1 cannot be relocalised from the plasma membrane. Thus we found that wild-type Cat1-GFP cells growing in nitrogen-starvation conditions is localised to the plasma membrane, particularly at the growing cell ends, as previously reported [12,26]. Interestingly, in an *any1R175C* background the fluorescence is internalised as recently reported [10], presumably to the Golgi, resulting in a canavanine-resistant phenotype (Fig 4). Any1/Arn1 over-expression causes a similar phenotype [26,27], so we assume that the *any1R175C* is a change-of-function mutation that causes hyperactivation of the Pub1-Any1 ubiquitin ligase. Modelling of the R175C mutation suggests that it enables Any1 to interact more strongly with Cat1, thus presumably promoting Cat1 ubiquitylation [9]. Other targets of Pub1 have been identified, such as Cdc25 [30], and it would be interesting to see if Any1R175C affects their ubiquitylation.

As noted, on PMG plates, deletion of both the *aat1*+ and *cat1*+ genes gives some resistance to canavanine, and the *any1R175C* mutation causes not only Cat1to be internalised but probably Aat1as well [27], thus a single mutation can inactivate at least two arginine transporters. Since two of the spontaneous *any1R175C* mutants, and indeed the original "*can1-1*" strain had additional mutations in genes implicated in canavanine resistance, it is likely that these additional mutations slightly enhance the canavanine-resistance phenotype of the *any1R175C* mutation. It is also of interest to consider why a mutation equivalent to R175C in the *S. cerevisiae* Any1-orthologue Art1/Ldb19 has not been described to confer canavanine resistance. This may reflect the fact that probability of a knock-out mutation in the single arginine transporter *CAN1* is higher than a specific point mutation in *ART1/LDB19*, whereas in *S. pombe* two arginine transporters need to be inactivated to provide canavanine resistance and this is less likely than the *any1R175C* mutation.

## Implications for use of canavanine in mutation rate assays in fission yeast

Although canavanine resistance is a simple selection for mutation rate determination in fission yeast and, from the panel of strains tested here, the mutation rate trends are broadly in line with FOA-resistance and *ade6-485* reversion, the restriction of most mutations to AnyR175C (C523T) means that this selectable mutation is not so useful for obtaining information about mutation spectra. Using different media, e.g. containing ammonium, might allow a wider selection of mutations to be selected for, for instance in the *cat1*+ gene.

## Supporting information

**S1 Fig. Effect of reducing the pH of PMG medium on canavanine-resistant colony appearance.** (A) 150 μl of a culture of strain 4355 (OD600 = 1) was spotted onto PMG + 80 μg/ml canavanine plates adjusted to the pHs shown. Plates were photographed after 5 days at 30˚C. (B) As (A) except WT, *any1R175C* and *pol2S298F* strains were used. For the *any1R175C* strain approximately 5 cells were plated.
(PDF)

**S2 Fig. Location of missense mutations in the amino-acid transporters Cat1 and Aat1.** Sequences aligned are *S. pombe* Cat1 and Aat1. The alignment shows that the missense mutations detected in some of the canavanine-resistant strains are conserved between Cat1 and Aat1. The alignment was generated by Praline [29].
(PDF)

**S3 Fig. Canavanine-resistant clones isolated from the hypermutation strain *pol2S298F* (3221) also have the *any1R175C* mutation.** Strains were genotyped by amplifying an *any1* gene fragment using oligos 1436 and 1437, followed by HpyCH4IV digestion. Lane 1, *any1R175C* control (strain 3647); lane 2, WT control (strain 2299); lanes 3–7, independent canavanine-resistant clones derived from hypermutating strain *pol2S298F* (3221).
(PDF)

**S1 Table. Strains used.**
(PDF)

**S2 Table. Oligos used.**
(PDF)

**S3 Table. Absolute mutation rates of Pol2 strains.**
(PDF)

**S1 Raw data.**
(PDF)

## Acknowledgments

We are very grateful to Drs Akio Nakashima, Kaoru Takegawa and Oliver Fleck for *S. pombe* strains. The original "*can1-1*" strain was provided by the YGRC/NBRP, Japan. We thank Peter Fantes and Li-Lin Du for discussions.

## Author Contributions

**Conceptualization:** Sophia Toumazou, Stephen E. Kearsey.

**Methodology:** Ellen Heitzer, Sibyl Bertrand.

**Project administration:** Chen-Chun Pai, Timothy C. Humphrey.

**Supervision:** Stephen E. Kearsey.

**Writing – original draft:** Stephen E. Kearsey.

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
