## [Decision Letter · Decision Letter 0]

15 Jul 2022

PONE-D-22-17741Using canavanine resistance to measure mutation rates in Schizosaccharomyces pombePLOS ONE

Dear Dr. Kearsey,

Thank you for submitting your manuscript to PLOS ONE. After careful consideration, we feel that it has merit but does not fully meet PLOS ONE’s publication criteria as it currently stands. Therefore, we invite you to submit a revised version of the manuscript that addresses the points raised during the review process.

The second reviewer has many concerns, both experimental and related to the presentation.  Many of the issues could be resolved by further analysis of the data and changes to the manuscript, but a few experiments may be necessary.  A revised manuscript should provide a full rebuttal to all points, concentrating on the points below:

[1] The reviewer considers that not enough recognition is given to previous work, and provides numerous references that should be cited (see suggestions throughout the report).

[2] The hypothesis of the authors regarding the cdc25 effect should be easily checked by WB to determine the protein level of Cdc25 in Any1+ and Any1-R175C.

[3] Fluctuation test data should be repeated using independent colonies.

Please also address the minor points raised by both reviewers.

We look forward to receiving your revised manuscript.

Kind regards,

Juan Mata, Ph.D.

Academic Editor

PLOS ONE

Journal Requirements:

"No"

5. Please ensure that you refer to Figure 5 in your text as, if accepted, production will need this reference to link the reader to the figure.

Reviewers' comments:

Reviewer's Responses to Questions

**Comments to the Author**

1. Is the manuscript technically sound, and do the data support the conclusions?

Reviewer #1: Yes

Reviewer #2: Partly

2. Has the statistical analysis been performed appropriately and rigorously? 

Reviewer #1: N/A

Reviewer #2: Yes

3. Have the authors made all data underlying the findings in their manuscript fully available?

Reviewer #1: Yes

Reviewer #2: Yes

4. Is the manuscript presented in an intelligible fashion and written in standard English?

Reviewer #1: Yes

Reviewer #2: Yes

5. Review Comments to the Author

Reviewer #1: Pai et al., ” Using canavanine resistance to measure mutation rates in Schizosaccharomyces

pombe”

In this paper, the authors investigate canavanine-resistant mutants of S. pombe as a tool to quantitate mutation rates. In S. cerevisiae, this is a main way to measure the rate of forward mutations. They find that, in contrast to S. cerevisiae where canavanine resistance largely arises from mutations in the CAN1 arginine transporter gene, in S. pombe all genome-sequenced resistant strains carry mutations in the any1+ gene, encoding a beta-arrestin and ubiquitin ligase adapter (homologous to S. cerevisiae ART1). Some isolates in addition show mutations in membrane transporter genes, but the authors conclude that the contribution to canavanine resistance from those are marginal. The authors then go on to show that in any1 mutants, the Cat1 membrane-bound nutrient transporter is absent from the plasma membrane and internalized. Specifically, the any1 point mutation (R175C) that is found in canavanine resistant strains, hits an evolutionarily conserved position that is close to a ubiquitylation site, necessary for regulation of the intracellular localization of the transporter.

A main conclusion is that the mutational spectrum in the any1+ gene isolated from canavanine-resistant mutants is limited to one single allele, and that this method therefore is not suitable for analyzing mutagenesis by DNA damaging agents or mutator alleles. As a measure of mutation rates to e.g. compare DNA damaging agents, canavanine resistance may work, although less efficiently than 5-FOA resistance.

The paper is focused and clearly written, some exceptions are listed below. The authors use adequate methods to discover the main mechanism for canavanine resistance in S. pombe, and describe the consequences of this for its use to measure mutation rates. The results are useful to the fission yeast community. I have only a few points that should be addressed:

In Fig. 1B, how are mutation rates actually represented? Taken literally, the Y axis would show rates relative to wt on a linear scale. The relative mutation rates for the mutants would then range from maybe 100 % (V412L on Ade) to 7000 % (S298F on 5-FOA) – is this correct? If so, all alleles would show hypermutation. Yet, the authors say that “V412L shows a low mutation rate, while the equivalent human mutation V411L causes hypermutation” (lines 154 -156). Please clarify what is meant by “low mutation rate” (= lower than wt or something else?), and explain the scale on the Y-axis.

Minor comments:

Line 148 - 150: The three measures of mutation rates show similar trends – why not show this by correlation coefficients? Or are the underlying numbers of mutant colonies counted too low to allow a statistical analysis?

The use of standard annotation for S. pombe should be used throughout the manuscript: lower case italics for mutants and mutations, same but followed by superscript “+” for genes and wt RNA; non-italics with initial upper case for proteins.

It would be interesting to hear the authors’ views on why the corresponding situation does not occur in S. cerevisiae: why are mutation found in the arginine transporter gene CAN1 in canavanine-resistant mutants, and not in ART1/LDB19, the S. cerevisiae ortholog of any1+?

Reviewer #2: Manuscript Number: PONE-D-22-17741 “Using canavanine resistance to measure mutation rates in Schizosaccharomyces pombe" by Chen-Chun Pai; Ellen Heitzer; Sibyl Bertrand; Sophia Toumazou; Timothy Humphrey; Stephen Kearsey.

In this study, Pai et al investigate the effect of mutant alleles of polymerase epsilon associated with cancers on mutagenesis using three assays available in S. pombe. In general, the paper consists of two parts: a small part describing the effect of the mutant alleles and a substantial part yet mostly reproducing other labs results dedicated to canavanine resistance in S. pombe. Although the impact of mutant alleles is potentially interesting, this study has multiple problems outlined below.

An effort should be made to substantiate certain findings (cdc25) and integrate accurately previous findings made by other labs. Specifically, most of the Any1R175C characterization by Pai et al. has already been reported. Cat1 internalization in Any1R175C was reported by Ait Saada et al. Canavanine resistance of vhc1, cat1, aat1 and cat1 aat1 deletions was reported by Aspuria and Tamanoi (PMID: 18219492), Ma et al. (PMID:23934889) and Ait Saada et al (PMID: 35639710). Two of those papers (PDIM 18219492, PDIM 35639710) are allegedly cited by Pai et al without giving credit to their findings. Instead of ignoring or overlooking other researchers' findings, the authors should make an effort to give more value to their real findings by giving more substance to their observations.

The only “new” results are figures 1B and 5. The rest is a reproduction of what is already known or simple alignments.

Abstract: “The any1R175C strain showed internalization of the Cat1 arginine transporter, explaining the canavanine resistance phenotype.” This has already been reported in Ait Saada et al, 2022 PlosOne and should not be presented as the main finding of this paper.

“it is likely that Cdc25 ubiquitylation and downregulation by Any1R175C-Pub1 delays mitotic entry.” This is only speculative, especially since the nature of the interaction between Any1-R175C and Pub1 is not known. Besides, the hypothesis of the authors regarding the cdc25 effect can be and should be easily checked by WB to determine the protein level of Cdc25 in Any1+ and Any1-R175C. In addition, Pub1-mediated Cdc25 ubiquitination can be assessed by WB in mts2ts mutant (Nefsky and Beach 1996 PMID: 8635463).

Fluctuation test data. The data are shown only relative to WT. What are the absolute rates? It is not clear what the error bars correspond to (CI, SD, SEM). In addition, the fluctuation test was performed from a single culture that was aliquoted into 12 wells of a 96-well plate, which means that the fluctuation test data are not obtained from independent clones. The reference cited by the authors (Lang and Murray 2008) reports using 10 independent clones. How many colonies were on average did grow on canavanine plates? It is also not clear whether the canavanine-resistant colonies obtained from the fluctuation test exhibit the any1-R175C mutation since it is not those clones that have been sequenced. Since POLE mutants are mutators by nature, there is no reason to expect only Any1-R175C mutation.

Section “Mechanism of canavanine resistance conferred by Any1R175C” There are no new findings in this result section. Cat1 internalization in Any1R175C has already been reported by Ait Saada et al. This reference is not cited even once in this result section.

Figure S1: The authors found that decreasing the pH in PMG media allows faster growth of canavanine-resistant cells. Is the effect of the pH specific to strain 4355? An appropriate strain control should be added (WT and can1-1).

Whole-genome sequencing. The strains sequenced are 2299 and 3938, WT and clr4Δ, respectively. It is not clear why the authors performed sequencing in clr4Δ background. Besides, none of the POLE mutants was included. Do the POLE mutants resistant to canavanine show any1-523C>T mutation and is C>T an expected mutational signature for the POLE mutants? Also, why POLE P286R was not include? In addition, the medium used to select for canavanine-resistant clones contains glutamate as a source of nitrogen, which is known to confer a stricter resistance compared to ammonium chloride. It is known that POLE mutants exhibit a mutator phenotype and that canavanine resistance in S. pombe can be very strong in certain double mutants. Since canavanine-resistant clones derived from POLE mutants have not been sequenced, it is not clear if the whole genome sequencing performed here provides a holistic view of the use of canavanine resistance to measure mutation rates.

Figure S2, vhc1 is not an ortholog of any arginine permease. It is not clear why it is part of the alignment. It is not even clear what is the purpose of this alignment per se. The secondary mutations identified in aat1 and cat1 are not shown to induce any phenotype related to canavanine resistance. The relevance of these mutations would be appreciated if the authors reproduce the single mutations and show that they behave like the corresponding single gene deletion. Using the appropriate medium, deletion of cat1 and aat1 (although to a lesser extent) have been shown to be resistant to canavanine (Aspuria and Tamanoi 2008).

Line 197. And figure 2. What is the relevance of the any1-R175 mutation being close to any2-K163? This later site is not very conserved between Any1 and Any2. Evidence shows that Any1 might be ubiquitinylated at K263 (Nakashima et al., 2014 PMID: 24876389), which corresponds to K486 in Art1 that has been shown to be ubiquitinated (Lin et al, 2008 PMID: 18976803).

Figure 3. and line 204 “To compare the canavanine resistance of an any-R175C strain with mutants defective in amino-acid transporters,” Vhc1 is not an amino acid transporter. The media used is PMG. It is known that glutamate leads to a higher uptake of arginine and explains why certain mutants are less resistant to canavanine in the presence of glutamate compared to ammonium chloride (Fantes and Creanor, 1984). In addition, all the deletion mutants used in this experiment have already been characterized for their canavanine resistance/sensitivity (Ma et al., 2013; Ait Saada et al., 2022; Aspuria and Tamanoi 2008).

Section “Implications for use of canavanine in mutation rate assays in fission yeast”. The authors conclude that only one kind of mutation can be selected for using their protocol which (i) makes the title of their paper misleading and (ii) the sensitivity of the assay questionable.

Minor comments

Line 97. no Δ should be in vhc1::hphMX6. The same applies to the other strain deletions in Table S1. Wild type should be written WT.

Line 104. Determination OF mutation rates by fluctuation analysis

Line 182. Replace “can1 gene” by “vhc1 gene previously called can1”.

Line 235. WT

Line 251. PMG

6. PLOS authors have the option to publish the peer review history of their article (what does this mean?). If published, this will include your full peer review and any attached files.

Reviewer #1: No

Reviewer #2: No

---

## [Author Response · Author response to Decision Letter 0]

1 Dec 2022

Comments to the Author

1. Is the manuscript technically sound, and do the data support the conclusions?

Reviewer #1: Yes

Reviewer #2: Partly

2. Has the statistical analysis been performed appropriately and rigorously? 

Reviewer #1: N/A

Reviewer #2: Yes

3. Have the authors made all data underlying the findings in their manuscript fully available?

Reviewer #1: Yes

Reviewer #2: Yes

4. Is the manuscript presented in an intelligible fashion and written in standard English?

Reviewer #1: Yes

Reviewer #2: Yes

5. Review Comments to the Author

Reviewer #1: Pai et al., ” Using canavanine resistance to measure mutation rates in Schizosaccharomyces

pombe”

In this paper, the authors investigate canavanine-resistant mutants of S. pombe as a tool to quantitate mutation rates. In S. cerevisiae, this is a main way to measure the rate of forward mutations. They find that, in contrast to S. cerevisiae where canavanine resistance largely arises from mutations in the CAN1 arginine transporter gene, in S. pombe all genome-sequenced resistant strains carry mutations in the any1+ gene, encoding a beta-arrestin and ubiquitin ligase adapter (homologous to S. cerevisiae ART1). Some isolates in addition show mutations in membrane transporter genes, but the authors conclude that the contribution to canavanine resistance from those are marginal. The authors then go on to show that in any1 mutants, the Cat1 membrane-bound nutrient transporter is absent from the plasma membrane and internalized. Specifically, the any1 point mutation (R175C) that is found in canavanine resistant strains, hits an evolutionarily conserved position that is close to a ubiquitylation site, necessary for regulation of the intracellular localization of the transporter.

A main conclusion is that the mutational spectrum in the any1+ gene isolated from canavanine-resistant mutants is limited to one single allele, and that this method therefore is not suitable for analyzing mutagenesis by DNA damaging agents or mutator alleles. As a measure of mutation rates to e.g. compare DNA damaging agents, canavanine resistance may work, although less efficiently than 5-FOA resistance.

The paper is focused and clearly written, some exceptions are listed below. The authors use adequate methods to discover the main mechanism for canavanine resistance in S. pombe, and describe the consequences of this for its use to measure mutation rates. The results are useful to the fission yeast community. I have only a few points that should be addressed:

In Fig. 1B, how are mutation rates actually represented? Taken literally, the Y axis would show rates relative to wt on a linear scale. The relative mutation rates for the mutants would then range from maybe 100 % (V412L on Ade) to 7000 % (S298F on 5-FOA) – is this correct? If so, all alleles would show hypermutation. Yet, the authors say that “V412L shows a low mutation rate, while the equivalent human mutation V411L causes hypermutation” (lines 154 -156). Please clarify what is meant by “low mutation rate” (= lower than wt or something else?), and explain the scale on the Y-axis.

The Y axis represents (mutation rate of mutant/mutation rate of wt) so a value of 1 indicates that the mutant has the same mutation rate as wt. The Pol2D276A E278A is devoid of 3’-5’ exonuclease activity, ie cannot proofread. Interestingly two of the mutants, Pol2S298F & Pol2S460F have higher mutation rates, so the mechanism of mutagenesis must involve something other than simple loss of proofreading. We define these two as hypermutators. We have clarified these points in the text and figure legend. 

Minor comments:

Line 148 - 150: The three measures of mutation rates show similar trends – why not show this by correlation coefficients? Or are the underlying numbers of mutant colonies counted too low to allow a statistical analysis?

We have included correlation coefficients in the legend to Fig 1B.

The use of standard annotation for S. pombe should be used throughout the manuscript: lower case italics for mutants and mutations, same but followed by superscript “+” for genes and wt RNA; non-italics with initial upper case for proteins.

These errors have been corrected.

It would be interesting to hear the authors’ views on why the corresponding situation does not occur in S. cerevisiae: why are mutation found in the arginine transporter gene CAN1 in canavanine-resistant mutants, and not in ART1/LDB19, the S. cerevisiae ortholog of any1+?

Assuming that the S. cerevisiae equivalent of any1R175C confers canavanine resistance, this difference may reflect the fact that probability of a knock out mutation in the single arginine transporter CAN1 is higher than a specific R175C mutation, whereas in pombe two arginine transporters need to be knocked out to provide CanR phenotype and this is less likely than the R175C mutation. 

Reviewer #2: Manuscript Number: PONE-D-22-17741 “Using canavanine resistance to measure mutation rates in Schizosaccharomyces pombe" by Chen-Chun Pai; Ellen Heitzer; Sibyl Bertrand; Sophia Toumazou; Timothy Humphrey; Stephen Kearsey.

In this study, Pai et al investigate the effect of mutant alleles of polymerase epsilon associated with cancers on mutagenesis using three assays available in S. pombe. In general, the paper consists of two parts: a small part describing the effect of the mutant alleles and a substantial part yet mostly reproducing other labs results dedicated to canavanine resistance in S. pombe. Although the impact of mutant alleles is potentially interesting, this study has multiple problems outlined below.

An effort should be made to substantiate certain findings (cdc25) and integrate accurately previous findings made by other labs. Specifically, most of the Any1R175C characterization by Pai et al. has already been reported. Cat1 internalization in Any1R175C was reported by Ait Saada et al. Canavanine resistance of vhc1, cat1, aat1 and cat1 aat1 deletions was reported by Aspuria and Tamanoi (PMID: 18219492), Ma et al. (PMID:23934889) and Ait Saada et al (PMID: 35639710). Two of those papers (PDIM 18219492, PDIM 35639710) are allegedly cited by Pai et al without giving credit to their findings. Instead of ignoring or overlooking other researchers' findings, the authors should make an effort to give more value to their real findings by giving more substance to their observations.

The only “new” results are figures 1B and 5. The rest is a reproduction of what is already known or simple alignments.

Although two very recently published papers show the importance of the R175C mutation in the original “can1-1” strain (Yang et al. 2022; Ait Saada et al. 2022), the focus of this paper is to determine the range of genes associated with spontaneous CanR mutants as generated in fluctuation analyses. We find at least with the selection used these mainly involve the R175C mutation which limits the usefulness of e.g. determining mutational spectra using canavanine resistance, unlike the situation in S. cerevisiae where a wide range of mutations in the CAN1 gene can be used to analyse mutational spectra.

Abstract: “The any1R175C strain showed internalization of the Cat1 arginine transporter, explaining the canavanine resistance phenotype.” This has already been reported in Ait Saada et al, 2022 PlosOne and should not be presented as the main finding of this paper.

The abstract has been modified to reflect this.

“it is likely that Cdc25 ubiquitylation and downregulation by Any1R175C-Pub1 delays mitotic entry.” This is only speculative, especially since the nature of the interaction between Any1-R175C and Pub1 is not known. Besides, the hypothesis of the authors regarding the cdc25 effect can be and should be easily checked by WB to determine the protein level of Cdc25 in Any1+ and Any1-R175C. 

We cannot see a difference in Cdc25-GFP levels comparing the any1R175C strain to wt by western blot. It is possible that Cdc25 ubiquitylation is altered but this does not affect the level of the protein. Alternatively, it is possible that the GFP tag interferes with Pub1-Any1 regulation of Cdc25. We have decided to remove discussion of Cdc25 from the paper as it is a peripheral point and further investigation would involve substantial work.

In addition, Pub1-mediated Cdc25 ubiquitination can be assessed by WB in mts2ts mutant (Nefsky and Beach 1996 PMID: 8635463).

Fluctuation test data. The data are shown only relative to WT. What are the absolute rates? 

We have included the absolute rates as a supplementary table.

It is not clear what the error bars correspond to (CI, SD, SEM).

This was explained in the Figure legend - they correspond to the range of data.

 In addition, the fluctuation test was performed from a single culture that was aliquoted into 12 wells of a 96-well plate, which means that the fluctuation test data are not obtained from independent clones. The reference cited by the authors (Lang and Murray 2008) reports using 10 independent clones. 

Each FA was conducted in triplicate or more from independent clones. We have clarified this point in the text.

How many colonies were on average did grow on canavanine plates? 

For wt approximately 0-1 colonies per spot; for the hypermutating variants 10-100.

It is also not clear whether the canavanine-resistant colonies obtained from the fluctuation test exhibit the any1-R175C mutation since it is not those clones that have been sequenced. Since POLE mutants are mutators by nature, there is no reason to expect only Any1-R175C mutation. 

We genotyped 5 independent canavanine-resistant colonies from strain pol2S298F and all had the any1-R175C mutation (Supplementary Figure 3).

Clearly a wider range of mutations will be found in the mutator strains, but the great majority of these will be passenger mutations unrelated to the CanR phenotype.

Section “Mechanism of canavanine resistance conferred by Any1R175C” There are no new findings in this result section. Cat1 internalization in Any1R175C has already been reported by Ait Saada et al. This reference is not cited even once in this result section.

Apologies for this oversight, this reference is now cited.

Figure S1: The authors found that decreasing the pH in PMG media allows faster growth of canavanine-resistant cells. Is the effect of the pH specific to strain 4355? An appropriate strain control should be added (WT and can1-1).

We have included controls to this figure.

Whole-genome sequencing. The strains sequenced are 2299 and 3938, WT and clr4Δ, respectively. It is not clear why the authors performed sequencing in clr4Δ background. 

We used a clr4Δ background for some of the sequencing in case epigenetic silencing (which would be abolished in clr4Δ, see Torres-Garcia S, et al. Nature. 2020;585(7825):453-8) was contributing to the CanR phenotype. However the CanR mutation rate did not differ between wt and clr4Δ strains, and sequencing the clr4Δ strains showed the same result as wt. We consider that discussion of this negative result would not contribute to the paper.

Besides, none of the POLE mutants was included. 

We did not include the POLE mutants for the sequencing as we were interested in mutations contributing to the CanR phenotype and the large number of passenger mutations in the mutator strains would confuse the analysis.

Do the POLE mutants resistant to canavanine show any1-523C>T mutation and is C>T an expected mutational signature for the POLE mutants? 

See point above. C>T mutations are not enhanced in S. pombe Pol2P287R compared to wt (Soriano et al. PLoS Genet. 2021;17(7))

Also, why POLE P286R was not include? 

We have published CanR data for this mutant (Soriano et al. PLoS Genet. 2021;17(7)) and did not want to duplicate it here.

In addition, the medium used to select for canavanine-resistant clones contains glutamate as a source of nitrogen, which is known to confer a stricter resistance compared to ammonium chloride. It is known that POLE mutants exhibit a mutator phenotype and that canavanine resistance in S. pombe can be very strong in certain double mutants. Since canavanine-resistant clones derived from POLE mutants have not been sequenced, it is not clear if the whole genome sequencing performed here provides a holistic view of the use of canavanine resistance to measure mutation rates.

See point above. The POLE mutants would not be a good choice for this sort of experiment. Due to passenger mutations in the POLE CanR strains it would be difficult to determine which mutations were relevant to the CanR phenotype. Clearly if different media were used with a wt strain, we might observe a wider spectrum of mutations, but we would consider this appropriate for a separate paper.

Figure S2, vhc1 is not an ortholog of any arginine permease. It is not clear why it is part of the alignment. 

We have deleted Vhc1 from the figure.

It is not even clear what is the purpose of this alignment per se. 

The purpose of the alignment is to show that the mutations identified in the sequence analysis affect amino acids conserved between Aat1 and Cat1 and are thus likely to have a phenotype. Since the double cat1� aat1� mutant is CanR, it is plausible that the single mutations contribute to the CanR phenotype in the any1R175C background.

The secondary mutations identified in aat1 and cat1 are not shown to induce any phenotype related to canavanine resistance. The relevance of these mutations would be appreciated if the authors reproduce the single mutations and show that they behave like the corresponding single gene deletion. 

The single gene deletions do not confer CanR at least on the selective medium we are using, so we would not expect point mutations to confer CanR on their own.

Using the appropriate medium, deletion of cat1 and aat1 (although to a lesser extent) have been shown to be resistant to canavanine (Aspuria and Tamanoi 2008).

This data was not shown in Aspuria and Tamanoi 2008 but referred to as unpublished – we show this result in Fig 3.

Line 197. And figure 2. What is the relevance of the any1-R175 mutation being close to any2-K163? This later site is not very conserved between Any1 and Any2. Evidence shows that Any1 might be ubiquitinylated at K263 (Nakashima et al., 2014 PMID: 24876389), which corresponds to K486 in Art1 that has been shown to be ubiquitinated (Lin et al, 2008 PMID: 18976803).

We are not meaning to imply any significance to the proximity of any1-R175C to any2-K163. There do not appear to be any lysines close to R175 in the Alphafold structure prediction. We have clarified this in the text.

Figure 3. and line 204 “To compare the canavanine resistance of an any-R175C strain with mutants defective in amino-acid transporters,” Vhc1 is not an amino acid transporter. 

We have included Vhc1 as it was originally designated Can1, but has shown in the figure, the deletion of this gene confers no CanR phenotype with the media used. We have clarified that it is not an amino-acid transporter.

The media used is PMG. It is known that glutamate leads to a higher uptake of arginine and explains why certain mutants are less resistant to canavanine in the presence of glutamate compared to ammonium chloride (Fantes and Creanor, 1984). In addition, all the deletion mutants used in this experiment have already been characterized for their canavanine resistance/sensitivity (Ma et al., 2013; Ait Saada et al., 2022; Aspuria and Tamanoi 2008).

Section “Implications for use of canavanine in mutation rate assays in fission yeast”. The authors conclude that only one kind of mutation can be selected for using their protocol which (i) makes the title of their paper misleading and (ii) the sensitivity of the assay questionable.

The data in Fig 1 show that carrying out fluctuation assays using canavanine gives a meaningful indication of the mutation rate using our conditions, and are similar to the Ade reversion assay, but the mutants obtained are not useful for analysis of mutational spectra. This point is clearly made in the text.

Minor comments

Line 97. no Δ should be in vhc1::hphMX6. The same applies to the other strain deletions in Table S1. Wild type should be written WT.

Many authors include � to indicate that the relevant gene is deleted rather than just tagged with the marker.

Line 104. Determination OF mutation rates by fluctuation analysis

Corrected

Line 182. Replace “can1 gene” by “vhc1 gene previously called can1”.

Corrected

Line 235. WT

Corrected

Line 251. PMG

Corrected

6. PLOS authors have the option to publish the peer review history of their article (what does this mean?). If published, this will include your full peer review and any attached files.

Do you want your identity to be public for this peer review? For information about this choice, including consent withdrawal, please see our Privacy Policy.

Reviewer #1: No

Reviewer #2: No

---

## [Decision Letter · Decision Letter 1]

19 Dec 2022

Using canavanine resistance to measure mutation rates in Schizosaccharomyces pombe

PONE-D-22-17741R1

Dear Dr. Kearsey,

We’re pleased to inform you that your manuscript has now been judged scientifically suitable for publication and will be formally accepted for publication once it meets all outstanding technical requirements.

Kind regards,

Juan Mata, Ph.D.

Academic Editor

PLOS ONE

Additional Editor Comments (optional):

Reviewers' comments:

Reviewer's Responses to Questions

**Comments to the Author**

1. If the authors have adequately addressed your comments raised in a previous round of review and you feel that this manuscript is now acceptable for publication, you may indicate that here to bypass the “Comments to the Author” section, enter your conflict of interest statement in the “Confidential to Editor” section, and submit your "Accept" recommendation.

Reviewer #1: All comments have been addressed

Reviewer #2: All comments have been addressed

2. Is the manuscript technically sound, and do the data support the conclusions?

Reviewer #1: Yes

Reviewer #2: Yes

3. Has the statistical analysis been performed appropriately and rigorously? 

Reviewer #1: Yes

Reviewer #2: Yes

4. Have the authors made all data underlying the findings in their manuscript fully available?

Reviewer #1: Yes

Reviewer #2: Yes

5. Is the manuscript presented in an intelligible fashion and written in standard English?

Reviewer #1: Yes

Reviewer #2: Yes

6. Review Comments to the Author

Reviewer #1: The authors have adquately addressed my concerns, mainly about Fig. 1. They have also made clear how the difference between fission and budding yeast in the number of arginine transporters can explain why canavanine resistance more often occurs through mutation in the CAN1 gene in S. cerevisiae, but in the any1+ gene in S. pombe.

Reviewer #2: The authors removed the cdc25 section and now introduced appropriate references. They also answered almost all my comments. As a result, the paper is significantly improved.

7. PLOS authors have the option to publish the peer review history of their article (what does this mean?). If published, this will include your full peer review and any attached files.

Reviewer #1: **Yes: **Per Sunnerhagen

Reviewer #2: No

---

## [Editor Report · Acceptance letter]

2 Jan 2023

PONE-D-22-17741R1 

Using canavanine resistance to measure mutation rates in *Schizosaccharomyces pombe*

Dear Dr. Kearsey:

I'm pleased to inform you that your manuscript has been deemed suitable for publication in PLOS ONE. Congratulations! Your manuscript is now with our production department. 

Kind regards, 

on behalf of

Prof Juan Mata 

Academic Editor

PLOS ONE